# HMG-CoA Reductase Inhibitor, Simvastatin Is Effective in Decreasing Degree of Myocarditis by Inhibiting Metalloproteinases Activation

**DOI:** 10.3390/biom11101415

**Published:** 2021-09-28

**Authors:** Monika Skrzypiec-Spring, Agnieszka Sapa-Wojciechowska, Katarzyna Haczkiewicz-Leśniak, Tomasz Piasecki, Joanna Kwiatkowska, Marzenna Podhorska-Okołów, Adam Szeląg

**Affiliations:** 1Department of Pharmacology, Wroclaw Medical University, 50-345 Wrocław, Poland; joanna.kwiatkowska@umed.wroc.pl (J.K.); adam.szelag@umed.wroc.pl (A.S.); 2Department of Medical Laboratory Diagnostics, Wroclaw Medical University, 50-013 Wrocław, Poland; agnieszka.sapa@umed.wroc.pl; 3Department of Ultrastructural Research, Wroclaw Medical University, 50-013 Wrocław, Poland; katarzyna.haczkiewicz@umed.wroc.pl (K.H.-L.); marzenna.poghorska-okolow@umed.wroc.pl (M.P.-O.); 4Department of Epizootiology and Clinic of Bird and Exotic Animals, Wroclaw University of Environmental and Life Sciences, 50-013 Wrocław, Poland; tomasz.piasecki@up.wroc.pl

**Keywords:** experimental autoimmune myocarditis, metalloproteinases, simvastatin

## Abstract

Background: Acute myocarditis often progresses to heart failure because there is no effective, etiology-targeted therapy of this disease. Simvastatin has been shown to be cardioprotective by decreasing matrix metalloproteinases’ (MMPs) activity. The study was designed to determine whether simvastatin inhibits MMPs activity, decreases the severity of inflammation and contractile dysfunction of the heart in experimental autoimmune myocarditis (EAM). Methods: Simvastatin (3 or 30 mg/kg/day) was given to experimental rats with EAM by gastric gavage for 21 days. Then transthoracic echocardiography was performed, MMPs activity and troponin I level were determined and tissue samples were assessed under a light and transmission electron microscope. Results: Hearts treated with simvastatin did not show left ventricular enlargement. As a result of EAM, there was an enhanced activation of MMP-9, which was significantly reduced in the high-dose simvastatin group compared to the low-dose group. It was accompanied by prevention of myofilaments degradation and reduction of severity of inflammation. Conclusions: The cardioprotective effects of simvastatin in the acute phase of EAM are, at least in part, due to its ability to decrease MMP-9 activity and subsequent decline in myofilaments degradation and suppression of inflammation. These effects were achieved in doses equivalent to therapeutic doses in humans.

## 1. Introduction

Myocarditis accounts for about 20% of cardiovascular diseases [1]. It may affect individuals of all ages, but it is most frequent in young people [1]. 2–42% cases of sudden cardiac death in young individuals are related to myocarditis [1]. The etiology of myocarditis often remains undetermined, but it may be induced by various infectious agents, drugs, toxins, and systemic diseases [2]. Viral infections are the most important causes of myocarditis [2]. In the myocardium of patients with myocarditis genomes of enterovirus, adenovirus influenza virus, human herpesvirus, Epstein-Barr virus, cytomegalovirus, and hepatitis C virus were detected. Recently, myocarditis cases have been documented in patients with severe acute respiratory syndrome coronavirus 2 (SARS-CoV-2) infection [3]. If no virus is identified and other causes are excluded, autoimmune myocarditis is presumed. Despite etiology, the treatment of myocarditis is mostly symptomatic. The prognosis of myocarditis patients varies according to the underlying etiology. In 25% of cases myocarditis progresses to dilated cardiomyopathy which has a poor prognosis [2]. It makes myocarditis one of the most important risk factors of heart failure in young individuals.

Therefore, the development of preventive and therapeutic approaches to myocarditis which may protect the heart from contractile dysfunction of the heart muscle and subsequent cardiomyopathy is of great importance. 

The lipid-lowering drugs, the 3-hydroxy-3-methylglutaryl coenzyme-A (HMG-CoA) reductase inhibitors (statins), in addition to their lipid-lowering action, possess several additional, lipid-independent, pleiotropic effects.

One of the pleiotropic actions of statins is their ability to decrease activation of matrix metalloproteinases (MMPs) [4,5,6,7,8,9,10,11,12,13,14]. Although MMPs are mainly known for their ability to degrade substrates in the extracellular matrix, it was also shown that they are localized intracellularly and contribute to the acute ischemia/reperfusion injury (I/Ri) of the heart muscle. In the settings of oxidative stress related to acute ischemia-reperfusion MMPs undergo activation and are responsible for cleaving troponin I, myosin light chain-1, α-actinin, and titin [15]. Enhanced activation of MMPs in heart tissue was observed also in experimental autoimmune myocarditis (EAM) [16,17,18,19]. It was proved that inhibition of MMPs activity by clarithromycin and carvedilol prevented impairment of cardiac contractility in EAM rats [16,18]. Moreover, it was shown that the inhibition of MMPs in EAM by carvedilol contributed to reduced degradation of troponin I and myofilaments [16].

Increasing evidence from animal studies suggests that statins reduce the severity of myocarditis [20,21,22,23,24,25,26,27,28,29,30]. The results of many studies demonstrated the ability of statins to decrease both histopathological severity of myocarditis and improve cardiac function through interference on the T-cell-mediated immune response [24,25,26,27,28,29]. It was also demonstrated that atorvastatin decreased inflammatory infiltration in EAM mouse hearts which was accompanied by suppression of the increase in tumor necrosis factor alpha (TNF-alpha) and interferon gamma (IFN-gamma) levels [30]. However, the interplay between reduction of severity of myocarditis and inhibition of MMPs activation in heart tissue by statin therapy has not been yet specifically studied.

Therefore, the present study was undertaken to verify the hypothesis that HMG-CoA reductase inhibitor, simvastatin, may decrease activation of metalloproteinases in heart tissue and protect the heart muscle form contractile dysfunction in EAM.

## 2. Materials and Methods

### 2.1. Animals for Active EAM

The EAM induction protocol was conducted on 20 Lewis strain female rats (6 to 8 weeks old) obtained from AnimaLab (Poznan, Poland). The animals were randomly divided into 4 groups and housed together in the same conditions.

### 2.2. Induction of Active EAM and Protocol of Experiment

The active EAM was inducted in accordance with a procedure described by Kodama et al [31]. In brief, rats received subcutaneous injections into one hind footpad with antigen-adjuvant emulsion prepared by emulsifying purified porcine cardiac myosin (Sigma-Aldrich, Poznan, Poland) with an equal volume of adjuvant complete Freund (Difco, Warszawa, Poland) supplemented with Mycobacterium tuberculosis strain H37Ra (Difco) to a final concentration of 5 mg/mL. 16 rats were injected with 0.1 mL of an emulsion into the footpad on days 0 and 7.

10 rats with induced EAM were divided into two groups (n = 5): high-dosage simvastatin group (30 mg/kg per day; EAM+S30) and low-dosage simvastatin group (3 mg/kg per day: EAM+S3). The drug was administrated orally by gastric gavage for 3 weeks, from day 0 to day 21 (Sigma-Aldrich). The next 6 rats with induced EAM (EAM, n = 6) and not immunized rats (C, n = 4) were given vehicle orally by gastric gavage for 3 weeks from day 0 to day 21.

### 2.3. Echocardiography

Transthoracic echocardiography was carried out on the 21st day by using the Esaote MyLab Delta imaging system with a 12 MHz transducer (Esaote, Maastricht, Holland). Two-dimensional imaging was performed in the parasternal short-axis view. After positioning the M-mode cursor at the level of the papillary muscles and interventricular septum, perpendicularly to the interventricular septum and the left ventricular (LV) posterior wall, parameters such as interventricular septum diameter (IVSDD), left ventricular end-diastolic diameter (LVEDD), left ventricular end-systolic diameter (LVESD) and left ventricular posterior wall end-diastolic diameter (LVPWDD) were measured. Three measurements of every parameter were done and averaged for evaluation for each rat. Fractional shortening (FS) of the left ventricle was calculated with the use of the formula: FS = [(LVEDD − LVESD)/LVEDD] × 100%.

### 2.4. Sample Collection

The study was approved by the Animal Ethics Committee of the Polish Academy of Science (decision No. 25/2012). All steps of the experiments have been conducted in a way to avoid animal suffering. On day 21, after echocardiography was performed, all animals were subjected to anesthesia with ketamine (10 mg/kg body weight) and then sacrificed by decapitation. The hearts were collected and divided into two parts. One part of each heart was rapidly deep-frozen and crushed into a powder at liquid nitrogen temperature and stored at −80 °C for subsequent biochemical analysis. The second part of each heart was processed for transmission electron microscopy (TEM) and hematoxylin-eosin (HE) staining. Prior to zymography and western blotting a proper amount of heart powder was mixed with homogenization buffer (50 mmol/L Tris-HCl pH 7.4, 150 mmol/L NaCl, 0.1% Triton X-100, and Protease Inhibitors Cocktail Set III (Sigma-Aldrich) to prepare 20% homogenates (*w*/*v*) with subsequent mechanical homogenization (3 times for 10 s on ice) and centrifugation (10,000× *g*, for 5 min at 4 °C). Protein content in supernatants was measured with Bradford Protein Assay (Bio-Rad, Warszawa, Poland) and bovine serum albumin (heat shock fraction, ≥98%, Sigma-Aldrich) served as a protein standard.

### 2.5. Measurement of MMP-2 and MMP-9 by Gelatine Zymography

The activity of gelatinases in samples was assessed using gelatine zymography on the basis of the technique described by Heussen and Dowdle [32] with necessary modifications. Samples were diluted to achieve 20 µg of protein in 20 μL, mixed with a proper amount of 4× Loading Buffer (BioRad), and applied to 7.5% acrylamide gels copolymerized with 2 mg of porcine gelatine (175 g Bloom, Sigma) per milliliter of gel. SDS-PAGE was run in BioRad Protean MiniCell at 120 V, 4 °C, then gels were washed with 2.5% Triton X-100 (3 × 20 min) to restore the activity of MMPs. Development of zymograms was performed for 18 h at 37 °C in the buffer containing 50 mmol/L Tris-HCl, 5 mmol/L CaCl_2_, 200 mmol/L NaCl, and 0.05% NaN3. Gels were stained with 0.5% Coomassie Brilliant Blue R-250, 30% methanol, 10% acetic acid for 2 h, and then destained in 30% methanol/10% acetic acid until the white bands on a dark blue background were visible. GS-800 Calibrated Densitometer with Quantity One 1-D analysis software v.4.6.9 (BioRad, Hercules, CA, USA) was used to scan and analyze gels and the relative MMPs activity was expressed in arbitrary units (AU) per microgram of total protein in a sample. MMPs were identified on the basis of their molecular weight in comparison to a capillary blood standard prepared according to Makowski and Ramsby [33].

### 2.6. Measurement of Troponin I by Western Blot

20 μL of each sample containing 1 μg/μL of protein were mixed with 4× Loading Buffer (BioRad) with the addition of 2-mercaptoethanol according to manufacturer’s instruction, and applied onto 10% SDS-PAGE gels. After completion of electrophoresis protein fractions were electroblotted onto a nitrocellulose membrane (by wet technique; 50 V, 30 min). The membrane was blocked with bovine serum albumin (heat shock fraction, ≥98%, Sigma) and incubated overnight at 4 °C with a primary monoclonal mouse antibody against cardiac troponin I (Thermo Fisher Scientific, Waltham, MA, USA) at 1:1000 dilution. After thorough washing, it was incubated with a secondary goat anti-mouse antibody conjugated with horseradish peroxidase (BioRad) at 1:1000 dilution for 1 h. Blots were developed by chemiluminescence assay (ClarityTM Western ECL Substrate, BioRad) and scanned using ChemiDocTM XRS+ System with Image LabTM Software v.5.2 (BioRad). Results were calculated from the standard curve. Rat cardiac troponin I (Advanced ImmunoChemical Inc., Long Beach, CA, USA) served as a standard.

### 2.7. Assessment of the Severity of the Inflammation in the Light Microscope (HE)

Collected samples of the myocardium were fixed in 4% buffered formalin (Chempur, Piekary Śląskie, Poland). Afterward, specimens were rinsed in running water, dehydrated in increasing concentrations of ethyl alcohol (Stanlab, Lublin, Poland), carried out through intermediate liquid Neo-Clear (Merck, Darmstadt, Germany), and embedded in paraffin blocks. The excessive paraffin was removed, exposing the surface of the sample with the embedded cardiac tissue. Subsequently, for every studied sample, four 600-nm-thick paraffin sections were prepared on a rotary microtome (Leica RM 2255, Nussloch, Germany). After deparaffinization, the sections were stained with Mayer’s hematoxylin and eosin, HE (Bio-Optica Milano, Italy) and sealed with a Euparal mounting agent (Roth, Mannheim, Germany). Next, the histological slides have dried for seven days at 37 °C. The evaluation of the severity of inflammation was performed for each studied group of animals: not immunized group without simvastatin, group with induced EAM without simvastatin, low-dosage simvastatin group EAM+S3, and high-dosage simvastatin group EAM+S30. The assessment of each histological slide was con-ducted by two independent researchers, without knowledge about the way of the treatment of the animals. The mononuclear cell infiltrates (MCIs) were been assessed for the whole cardiac muscle section under a light microscope BX53 (Olympus, Tokyo, Japan) with magnification 20. Then, the inflammatory status in tissue sample was classified with the usage of the following scale developed by Godsel et al. [34]: normal (without any signs of inflammation); mild (≤10% of the whole cardiac tissue section was occupied by MCIs), moderate (>10% and ≤25% of the whole cardiac tissue section was occupied by MCIs), and severe (>25% of the whole cardiac tissue section was occupied by MCIs).The HE-stained cardiac tissue has been scanned with the usage of a Pannoramic Midi II histological scanner (3D HISTECH Ltd., Budapest, Hungary). The images were captured with the Pannoramic Viewer (3D HISTECH Ltd., RRID: SCR_014424, Budapest, Hungary) Software.

### 2.8. Assessment of the Myofibrils in the Transmission Electron Microscope (TEM)

The cardiac samples were fixed in 2.5% glutaraldehyde (Serva Electrophoresis, Heidelberg, Germany) diluted in cacodylate buffer (0.2 M, pH 7.4, Serva Electrophoresis). Then the fixative was washed out four times in the cacodylate buffer for 15 min. Subsequently, a solution of osmium tetroxide (Serva) was used as a secondary fixative for 1 h, followed by rinsing the samples in the 0.1 M cacodylate buffer 4 × 5 min. Dehydration of the specimens was carried out in the increasing concentrations of ethyl alcohol up to three changes of pure acetone (Chempur). Then the material was transferred to a mixture of acetone-epon. Afterwards, samples were embedded in epoxy resin Epon 812 (Serva Electrophoresis). Epon blocks were cut on a Power Tome XL ultramicrotome (RMC, Tucson, AZ, USA). Semithin sections, 600-nm-thick and ultrathin sections, 70-nm-thick were prepared using the histo diamond knife (Diatome, Nidau, Switzerland). The dried semithin sections were stained with toluidine blue (Serva Electrophoresis). Ultrathin sections were applied on the rhodium–copper grids (Maxta form, 200 mesh, Ted Pella), allowed to air dry for 24 h, and finally counterstained with uranyless EM stain and lead citrate (Science Services, München, Germany). The ultrathin sections were observed in a transmission electron microscope TEM JEM-1011 (Jeol, Tokyo, Japan), in order to determine the structure of myofilaments in cardiac muscle. The electronograms were collected using TEM imaging platform iTEM1233 equipped with the Morada Camera (Olympus Soft Imaging Solutions, EMSIS GmbH, Münster, Germany), under magnification ranging from 3 to 15 K.

### 2.9. Statistical Analysis

GraphPad Prism 7.0 was used for the statistical analysis (La Jolla, CA, USA). All data are expressed as mean ± SEM. The statistical analysis was performed with ANOVA and Kruskal-Wallis one-way analysis of variance. The post-hoc analysis was done using the Dunn test with Bonferroni correction. *p*-values < 0.05 were considered statistically significant.

## 3. Results

### 3.1. Influence of Simvastatin on Left Ventricular Enlargement in Rat Hearts Subjected to EAM

Echocardiographic results showed a significant (0.53 ± 0.02 vs. 0.64 ± 0.11, n = 5, *p* < 0.05) increase in LVEDD in EAM rats in comparison with the C group (Figure 1). Both groups of simvastatin treated rats showed no significant difference with the C group (Figure 1).

### 3.2. Inhibition of MMP-9 Activity by Simvastatin in Rat Hearts Subjected to EAM

As a result of EAM, there was an enhanced activation of MMP-9 especially in EAM+S3 group, which was abolished by high dose of simvastatin—it was a significant reduction of MMP-9 activity in the EAM+S30 group in comparison with the EAM+S3 group (58.37 ± 33.63 AU/μg vs. 10.30 ± 8.72 AU/μg, n = 5 *p* < 0.05) (Figure 2).

### 3.3. Simvastatin Has no Significant Effect on MMP-2 Activity

Simvastatin in both doses tested has no significant effect on MMP-2 activation in rat hearts subjected to EAM (data not shown).

### 3.4. Troponin Level in Simvastatin Treated Rat Hearts Subjected to EAM

Western blot showed no significant differences in troponin levels between groups, however, the mean troponin level was the highest in the EAM+S30 group (13.75 ± 2.44 ng/μL vs. 13.72 ± 2.33, 10.36 ± 4.11, 9.85 ± 1.09 ng/μL in C, EAM, and EAM+S3 groups respectively (Figure 3).

### 3.5. Reduction of the Severity of Inflammation by Simvastatin in Rat Heart Subjected to EAM

There was no inflammation observed in the control group (group C) (Figure 4).

The most severe inflammatory cell infiltrations in the perimysium and between cardiomyocytes were observed in the untreated rats with induced EAM (group EAM). The inflammatory cells formed thick aggregations. In turn, in the EAM+S30 group (high-dosage simvastatin) and in the EAM+S3 group were observed less severe, diffuse mononuclear cells infiltrate. The largest severity of inflammation occurred in the subendocardial area. Inflammatory infiltrates were composed of neutrophils, lymphocytes, macrophages, and plasma cells. The multinuclear giant cells were also seen, mostly in the rats of group EAM and EAM+S3, and were delimited to the most severe infiltration area (Figure 5).

In EAM+S30 group a severity of inflammation was significantly lower in comparison with EAM rats (3.17 ± 0.49 vs. 3.9 ± 0.12, n = 5, *p* < 0.05) (Figure 6).

### 3.6. Protection of Myofibrils Degradation by Simvastatin in Rat Hearts Subjected to EAM

The cardiac muscle in the not immunized rats demonstrated normal cross striations and ultrastructure of myofibrils with well-preserved, plentiful mitochondria (Figure 4). The most severe degradation of myofibrils was in rats of the non-treated EAM group. In the simvastatin treated animals, the changes in myofibrils were more pronounced mainly in group EAM+S3 while in rats of EAM+S30 were less distinct. In the case of both treated groups, the degradation of myofibrils was focal rather than diffuse. Intercalated discs in group EAM+S3 in the regions of desmosomes were disrupted, and myofilaments were detached. The swollen mitochondria were also observed. On the other hand, fewer pathological changes were observed for the EAM+S30 animals. In the predominant number of samples, the structure of the contractile apparatus was normal (Figure 7).

## 4. Discussion

Although the etiology of myocarditis includes a large variety of infectious agents, mainly viral, including SARS-CoV-2, as well as drugs, toxins, and systemic diseases, etiology-targeted therapy in most cases is not available [2,3]. This is on account of the fact that there is still no approved antiviral therapy for the treatment of most viral infections including SARS-CoV2 infection. Therefore, the core principles of treatment in acute myocarditis are the treatment of arrhythmia, and preventing of the heart failure with diuretics, angiotensin-converting enzyme inhibitor, or angiotensin receptor blockade, beta-adrenergic blockade, and aldosterone antagonists [2].

As many patients may present persistent heart failure symptoms despite standard optimal management, additional treatment strategies which could prevent contractile dysfunction of the heart are being sought. Immunomodulatory therapy including some anti-viral therapies, high dose intravenous immunoglobulin, immunoadsorption, or immunosuppressive therapy which modulate the immune, and inflammatory response are new therapeutic options [35,36,37,38,39,40,41]. Currently some clinical trials with their use like “Anakinra Versus Placebo for the Treatment of Acute Myocarditis”, “Clinical Assessment of New Treatment Regimen for Adult Fulminant Myocarditis”, “Study to Evaluate the Efficacy of Immunosuppression in Myocarditis or Inflammatory Cardiomyopathy” are underway [42].

Other interesting and promising treatment options may be strategies of preventing oxidative stress-related destruction of contractile proteins [16,18]. Oxidative stress related to acute ischemia-reperfusion injury of the heart muscle leads to intracellular activation of MMPs and subsequent degradation of troponin I, myosin light chain-1, α-actinin, and titin [15]. Increased activation of MMPs such as MMP-3 and MMP-9 were also observed in experimental mouse autoimmune myocarditis [17]. Hishikari et al. showed that MMPs inhibitor clarithromycin attenuates myocarditis and prevents subsequent impairment of cardiac function by suppressing MMP-9 [18]. Guttierez et al. showed increased enzymatic activity for MMP-2 and MMP-9 in heart tissue during the acute phase of experimental Trypanosoma cruzi infection and significant decrease of heart inflammation, delayed peak in parasitemia, and improved survival rates as a result of inhibition of MMPs activity (19). In our previous study, we also showed that experimental autoimmune myocarditis in rats proceeded with an enhanced activation of MMPs in heart tissue and that inhibition of MMP-2 activity by carvedilol resulted in the reduction of troponin and myofilaments degradation and subsequent prevention of dysfunction of the mechanical function of the heart [16].

In the present study, we found that in experimental autoimmune myocarditis in rats the use of a high dose of simvastatin (30 mg/kg/day) improved heart function and decreased degree of inflammation. These data correspond with several prior studies on statins in EAM [24,25,26,27,28,29,30]. The results of these studies, in which different statins were used (atorvastatin, fluvastatin, rosuvastatin, and pitavastatin), demonstrated the ability of this class of drugs to decrease severity of the disease and that effectiveness of statins was related to a sequential interference on the T-cell-mediated immune response [25,26,27,28,29,30]. Influence of simvastatin on EAM was investigated by Wu et al. They demonstrated that simvastatin ameliorated EAM through the inhibition of cross-talk between lymphocytes and antigen presenting cells [27].

In our study we demonstrated that simvastatin reduced mechanical dysfunction of heart muscle and decreased severity of inflammation along with attenuation of MMP-9 activity and myofilaments degradation in hearts subjected to EAM. These data support the hypothesis that beneficial effects of simvastatin on EAM are associated not only with the previously reported influence on immune response, but also include a concomitant suppressive action on MMP-9 activity, leading to subsequent reduction in myofilaments degradation and prevention of mechanical dysfunction of the heart.

To our knowledge, this is the first demonstration that HMG-CoA reductase inhibitors such as simvastatin may be effective in decreasing the degree of myocarditis by inhibiting MMPs activation, thus suggesting the interest for clinical studies.

In animal studies, a dose of simvastatin of 50 mg/kg per day produced the greatest incidence of main side effects such as hepatotoxicity, myopathy or cataracts while a dose of 10 mg/kg per day was not associated with any evidence of side effects [43,44]. Then, the range of human therapeutic doses of simvastatin can be compared to the one between the no-effect dose (10 mg/kg per day) to the effect dose (50 mg/kg per day) in animals [43,44]. In this context, it is important to point out that the dose of 30–40 mg/kg/day, which was effective in our study, is, therefore, an equivalent of the average dose used in humans with a low risk of systemic side effects.

## 5. Conclusions

This study shows for the first time that simvastatin reduces the severity of EAM in rats and this effect includes improved cardiac function and is associated with suppression of MMP-9 activation along with reduction of myocardial inflammation and myofilament destruction. These beneficial effects of simvastatin in EAM are achieved in doses equivalent to average therapeutic doses in humans. Thus, therapy with simvastatin could be a potentially novel and encouraging approach to preventing heart failure in myocarditis in humans, although further studies are needed to clarify and evaluate its clinical usefulness.

## Figures and Tables

**Figure 1 biomolecules-11-01415-f001:**
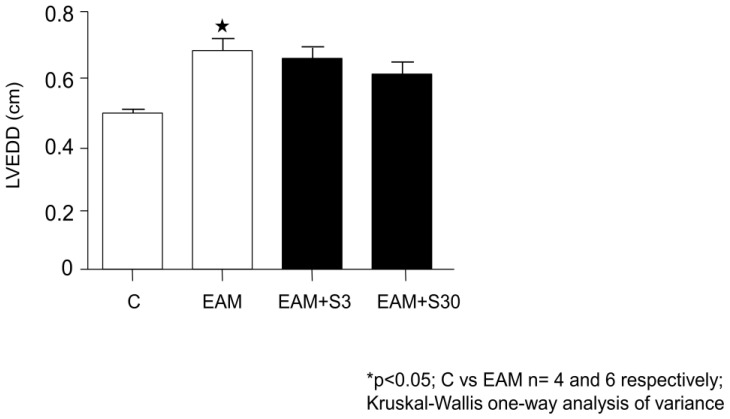
Effect of simvastatin on left ventricular end-diastolic diameter (LVEDD). C indicates not immunized group without simvastatin, EAM—group with induced EAM without simvastatin; EAM+S3—low-dosage simvastatin EAM group (3 mg/kg/d); EAM+S30—high-dosage simvastatin EAM group (30 mg/kg/d).

**Figure 2 biomolecules-11-01415-f002:**
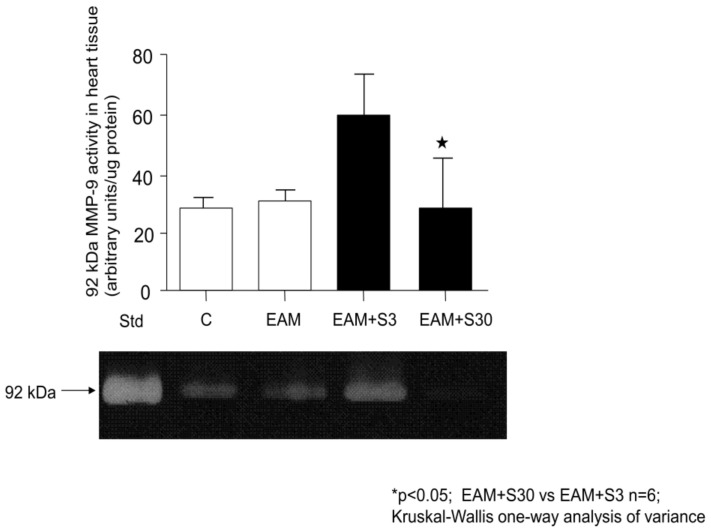
92-kDa MMP-9 activities in heart tissue and representative zymogram showing gelatinolytic activities in heart tissue. C indicates not immunized group without simvastatin; EAM—group with induced EAM without simvastatin; EAM+S3—low-dosage simvastatin EAM group (3 mg/kg/d); EAM+S30—high-dosage simvastatin EAM group (30 mg/kg/d); 92 kDa indicates matrix metalloproteinase 9.

**Figure 3 biomolecules-11-01415-f003:**
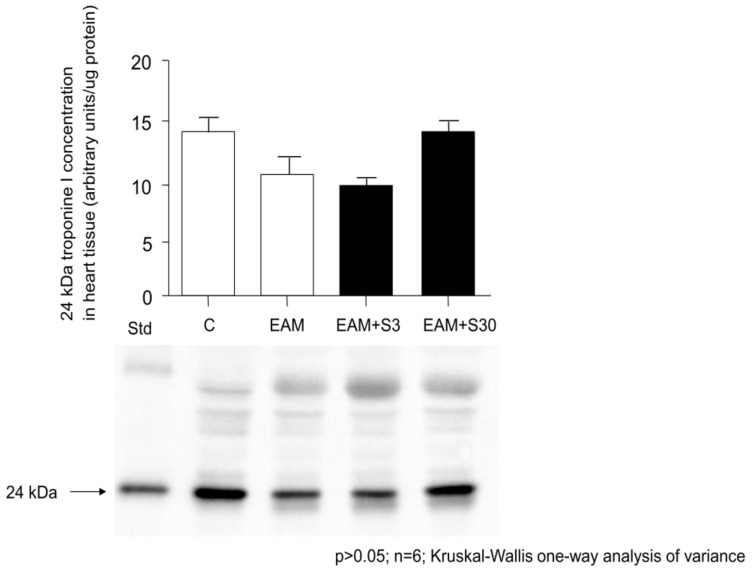
24-kDa troponin I (TnI) levels in heart tissue and representative TnI protein quantity in heart homogenates determined by Western blot. C indicates not immunized group without simvastatin; EAM—group with induced EAM without simvastatin; EAM+S3—low-dosage simvastatin EAM group (3 mg/kg/d); EAM+S30—high-dosage simvastatin EAM group (30 mg/kg/d); 24 kDa indicates TnI.

**Figure 4 biomolecules-11-01415-f004:**
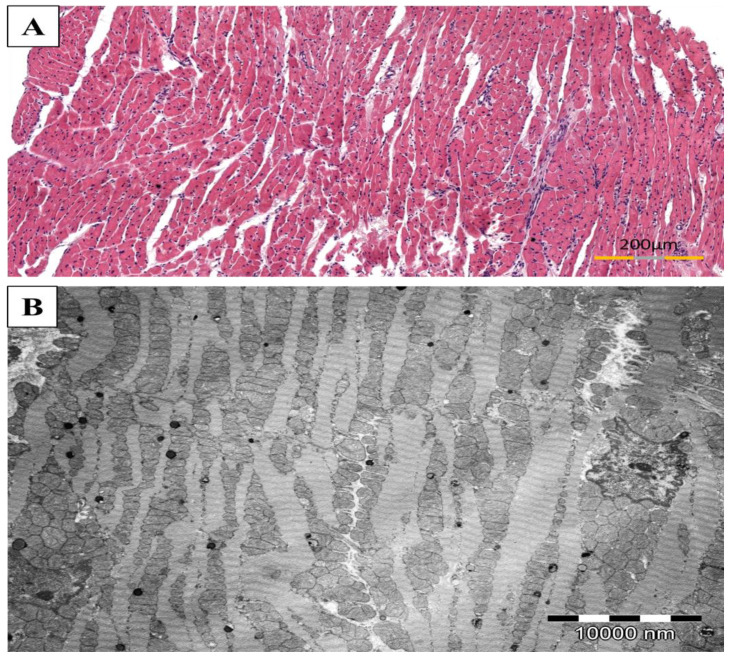
The representative images of cardiac muscle from not immunized rats without simvastatin (group C). The healthy heart without mononuclear cell infiltrates ((**A**), HE stain) and a typical ultrastructure of heart with proper cross striations and abundant mitochondria ((**B**), TEM).

**Figure 5 biomolecules-11-01415-f005:**
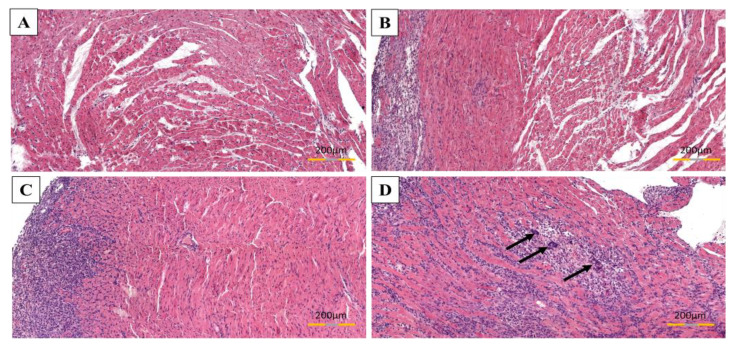
The acuteness of inflammation in rats’ cardiac muscle subjected to EAM. Area of heart without inflammation (**A**), a mild degree of inflammation (**B**), a moderate degree of inflammation (**C**), severe degree of inflammation; arrows indicate multinucleated giant cells (**D**). HE stain.

**Figure 6 biomolecules-11-01415-f006:**
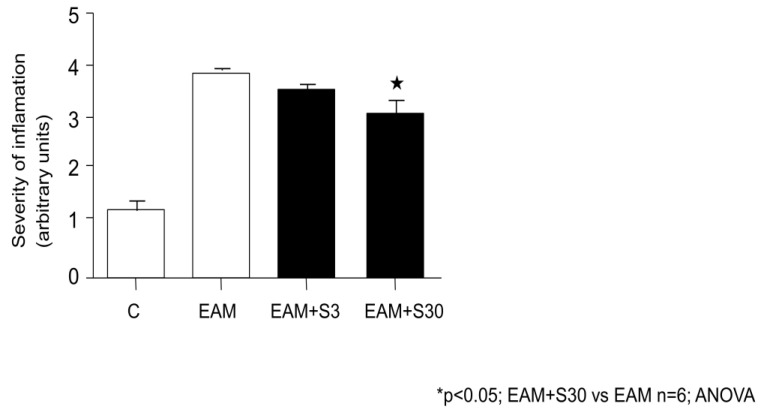
Effect of simvastatin on severity of inflammation. C indicates not immunized group without simvastatin, EAM—group with induced EAM without simvastatin; EAM+S3—low-dosage simvastatin EAM group (3 mg/kg/d); EAM+S30—high-dosage simvastatin EAM group (30 mg/kg/d).

**Figure 7 biomolecules-11-01415-f007:**
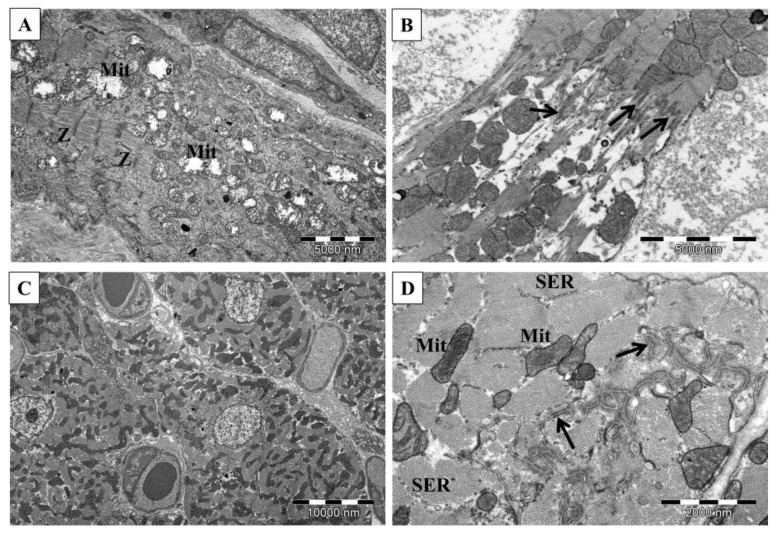
Ultrastructure of the cardiac muscle in rats subjected to EAM ((**A**,**B**)—EAM S3; (**C**,**D**)—EAM S30). Swollen mitochondria and substantial loss of myofilaments (**A**), myofilaments connected with the intercalated disc show disarrangement (**B**), normal heart (**C**), the well-organized structure of intercalated disc (**D**). Mit—mitochondria, Z—Z lines, arrows—intercalated disc, SER—sarcoplasmic reticulum. TEM.

## Data Availability

The data used to support the findings of this study are included within the article and are available from the corresponding author.

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
