# Peer review of "HMG-CoA Reductase Inhibitor, Simvastatin Is Effective in Decreasing Degree of Myocarditis by Inhibiting Metalloproteinases Activation"

_biomolecules, 2021, doi:10.3390/biom11101415_

Round 1
Reviewer 1 Report
The topic is interesting and necessary however introduction and discussion is weak. Literature is not up to date and original studies are not cited – instead there are review articles cited.
Abstract
Line 14 From this sentence one may understand that if there was such targeted therapy of myocarditis then myocarditis would not preceded HF what is not true. It rather should be written that HF would not occur if there was a target therapy of myocarditis,
Introduction
Line 34 cardiovascular diseases or cardiovascular events not cardiovascular disease events
Line 35 English editing needed
Lines 37-41 lack citation
Line 42 after infection or as a cause of infection ?
Line 43 citation is related to SARS-COV-2 but it is from the year 2013 when there was no SARS-Co-V2
Line 55 various animal studies are mentioned but neither one is cited
Line 58 what is I/Ri ? it is not explained
Line 56-61 are difficult to understand
Line 66-67- review articles are cited not the ones that are directly related to the effect caused by statins. This mechanism should be wider explained.
Line 70 is ischemia – reperfusion injury present in myocarditis ? if so it should be explained
Line 71 HMG-CoA- is not explained
Line 73 only in the setting of oxidative stress ?
discussion
Line 268 lack citation
Line 275 the citation is not adequate, no studies which are underway are cited
Line 277 lack citation
Line 292-294 is an aim not discussion
Line 302-303 what side effects ? which animal studies ? e review article is cited
Line 307 – what is average dose of simvastatin in humans ?
Author Response
Reviewer #1 comments:
Comments and Suggestions for Authors
The topic is interesting and necessary however introduction and discussion is weak. Literature is not up to date and original studies are not cited – instead there are review articles cited.
Response: We greatly appreciate Reviewer #1 comments. Changes suggested have been done and they have significantly improved the introduction and discussion sections. These changes required the addition of 30 new citations which are now added to reference list on pages 14–16 lines 441–475, 503–527, 537–561.
The numbers of subsequent references needed to be changed as well as citations numbers in the text.
The responses to the specific comments are as follows:
Abstract
Line 14 From this sentence one may understand that if there was such targeted therapy of myocarditis then myocarditis would not preceded HF what is not true. It rather should be written that HF would not occur if there was a target therapy of myocarditis.
Response:
The text on page 1, line 14 was changed according to Reviewer #1 suggestions:
„Acute myocarditis often progresses to heart failure because there is no effective, etiology-targeted therapy of this disease“.
Introduction
Line 34 cardiovascular diseases or cardiovascular events not cardiovascular disease events
Response:
As per Reviewer #1 suggestion the sentence was changed as follows (page 1 line 34) :
„Myocarditis accounts for about 20% of cardiovascular diseases [1].“
Line 35 English editing needed
Response:
In response to Reviewer #1 comment the text undewent English revision and was changed as follows (page 1, lines 34–36):
„It may affect individuals of all ages, but it is most frequent in young people [1]. 2-42% cases of sudden cardiac death in young individuals are related to myocarditis [1].“
Lines 37-41 lack citation
Response:
The lacking citations were added to the text on page 1 and 2, lines 38-43:
„The etiology of myocarditis often remains undetermined, but it may be induced by various infectious agents, drugs, toxins, and systemic diseases [2]. Viral infections are the most important causes of myocarditis [2].“
Line 42 after infection or as a cause of infection ?
Response:
Pathophysiology of myocarditis related with SARS-COV-2 infection is a combination of the direct viral damage of cardiomyocytes and the immune response to viral infection. Therefore the symptoms emerge in acute phase of the infection. Therefore the sentence was changed as follows (page 2, line 42–43):
„Recently, myocarditis cases have been documented in patients with severe acute respiratory syndrome coronavirus 2 (SARS-CoV-2) infection [3].“
Line 43 citation is related to SARS-COV-2 but it is from the year 2013 when there was no SARS-Co-V2
Response:
The adequate reference was added on page 2 line 43 and to reference list on page 13, line 410.
Line 55 various animal studies are mentioned but neither one is cited
Response:
As per Reviewer #1 suggestion new citations were added to the text on page 2, line 56 and to reference list on page 13, lines 414-444:
„One of the pleiotropic actions of statins is their ability to decrease activation of matrix metalloproteinases (MMPs) [4-14].“
Line 58 what is I/Ri ? it is not explained
Response:
The acronym I/Ri was indicated in full text on page 2 line 58:
„Although MMPs are mainly known for their ability to degrade substrates in the extracellular matrix, it was also shown that they are localized intracellularly and contribute to the acute ischemia/reperfusion injury (I/Ri) of the heart muscle. In the settings of oxidative stress related to acute ischemia-reperfusion MMPs undergo activation and are responsible for cleaving troponin I, myosin light chain-1, α-actinin, and titin [15].“
Line 56-61 are difficult to understand
Response:
To make tthis paragraph more clear to the reader we changed the text on page 2 lines 61-66 as follows:
„Enhanced activation of MMPs in heart tissue was observed also in experimental autoimmune myocarditis (EAM) (16-19). It was proved that inhibition of MMPs activity by clarithromycin and carvedilol prevented impairment of cardiac contractility in EAM rats [16, 18]. Moreover, it was shown that the inhibition of MMPs in EAM by carvedilol contributed to reduced degradation of troponin I and myofilaments [16].
Line 66-67- review articles are cited not the ones that are directly related to the effect caused by statins. This mechanism should be wider explained.
Response:
As per Reviewer #1 suggestion the primary sources were cited in the text on page 2 lines 67–73 and added to reference list on page 14, lines 461–487, and the text was changed as follows:
„Increasing evidence from animal studies suggests that statins reduce the severity of myocarditis [20-30]. The results of many studies demonstrated the ability of statins to decrease both histopathological severity of myocarditis and improve cardiac function through interference on the T-cell-mediated immune response [24-29]. It was also demonstrated that atorvastatin decreased inflammatory infiltration in EAM mouse hearts which was accompanied by suppression of the increase in tumor necrosis factor alpha (TNF-alpha) and interferon gamma (IFN-gamma) levels [30].“
Line 70 is ischemia – reperfusion injury present in myocarditis ? if so it should be explained
We thank Reviewer #1 for this comment. The ischemia – reperfusion injury could possibly be present in myocarditis. It would require disproportionately long explenation. Because this is not crucial to the manuscript we decided not to discuss the problem and the sentence on page 2, lines 76–78 were changed as follows:
„Therefore, the present study was undertaken to verify the hypothesis that HMG-CoA reductase inhibitor, simvastatin, may decrease activation of metalloproteinases in heart tissue and protect the heart muscle form contractile dysfunction in EAM.“
Line 71 HMG-CoA- is not explained
Response:
The acronym HMG-CoA was indicated in full text on page 2 lines 52–53:
„The lipid-lowering drugs, the 3-hydroxy-3-methylglutaryl coenzyme-A (HMG-CoA) reductase inhibitors (statins), in addition to their lipid-lowering action, possess several additional, lipid-independent, pleiotropic effects.“
Line 73 only in the setting of oxidative stress ?
Response:
The sentence from page 1 line 73 was changed for the reason explained above.
discussion
Line 268 lack citation
Response:
The lacking citations were added to the text on page 10, lines 294–296:
„Although the etiology of myocarditis includes a large variety of infectious agents, mainly viral, including SARS-CoV-2, as well as drugs, toxins, and systemic diseases, etiology-targeted therapy in most cases is not available [2-3].“
Line 275 the citation is not adequate, no studies which are underway are cited
Response:
As per Reviewer #1 suggestion the adequate reference was added on page 10, lines 307–311 and to the reference list on page 14, lines xxx and examples of studies which are underway were added to the text on page 10, line 516:
„Currently some clinical trials with their use like “Anakinra Versus Placebo for the Treatment of Acute Myocarditis”, “Clinical Assessment of New Treatment Regimen for Adult Fulminant Myocarditis”, “Study to Evaluate the Efficacy of Immunosuppression in Myocarditis or Inflammatory Cardiomyopathy” are underway [42].“
Line 277 lack citation
Response:
The lacking citations were added to the text on page 10, line 312–313:
„Other interesting and promising treatment options may be strategies of preventing oxidative stress-related destruction of contractile proteins [16, 18].“
Line 292-294 is an aim not discussion
Response:
We thank Reviewer #1 for this comment. The paragraph was removed from the manuscript.
Line 302-303 what side effects ? which animal studies ? a review article is cited
Response:
In response to Reviewer #1 comment and the text on page 11, lines 348–350 were changed as follows:
„In animal studies, a dose of simvastatin of 50 mg/kg per day produced the greatest incidence of main side effects such as hepatotoxicity, myopathy or cataracts while a dose of 10 mg/kg per day was not associated with any evidence of side effects [43, 44].“
The primary source was cited on page 11 line 350 and 353 and added to reference list on pages 14–15, lines 517–521.
Line 307 – what is average dose of simvastatin in humans ?
Response:
In response to Reviewer #1 comment the average doses of simvastatin in humans were added to the text on page11, line 353–355:
„In this context, it is important to point out that the dose of 30-40 mg/kg/day, which was effective in our study, is, therefore, an equivalent of the average dose used in humans with a low risk of systemic side effects.“
Reviewer 2 Report
The authors investigate a possible pleiotropic effect of statins on myocarditis. The article presents s good explanation in methods and results are well presented.
However, there are some issues that need to be addressed
- acronyms used were not indicated in full text the first time they were nominated. It happens most in introduction. I suggest also to consider separately abstract and main text
- authors indicated they evaluated some CD positivity in tissue sample. However, no data were reported about each CD named. I suggest to indicate them to increase the value of the manuscript
- I did not understand how they classified inflammatory status in tissue sample. I suggest to be more clear and more specific in explanation
- Discussion is too short in evaluating similarities and differences to other possible use of statins as pleiotropic effect. In fact, evidences were already reported on the effects of statins during infections. Please address
Author Response
Reviewer #2 comments:
Comments and Suggestions for Authors
The authors investigate a possible pleiotropic effect of statins on myocarditis. The article presents good explanation in methods and results are well presented.
However, there are some issues that need to be addressed
Response:
We thank Reviewer #2 for appreciative comments of our work. We have made the suggested changes and it significantly improved the manuscript.
The responses to the specific comments are as follows:
acronyms used were not indicated in full text the first time they were nominated. It happens most in introduction. I suggest also to consider separately abstract and main text
Response:
As per Reviewer #2 suggestion all acronyms used in the manuscript were indicated in full text the first time they were nominated separately in abstract and in main text. It required changes to the text on pages 1–2, lines 16, 52, 55–56, 58, and 62.
Authors indicated they evaluated some CD positivity in tissue sample. However, no data were reported about each CD named. I suggest to indicate them to increase the value of the manuscript
Response:
Unfortunately we did not performed immunohistochemical imaging of heart tissue. We thank the Reviewer #2 for an interesting suggestion that we will consider for a future study.
I did not understand how they classified inflammatory status in tissue sample. I suggest to be more clear and more specific in explanation
Response:
We thank the Reviewer #2 very much for this tip and valuable comments. To make it more clear to the reader, the more specific explanation was included in the section „2.7 Assessment of the severity of the inflammation in the light microscope (HE)“. (page 4, lines 160–186).
„Collected samples of the myocardium were fixed in 4% buffered formalin (Chempur, Piekary ÅšlÄ…skie, Poland). Afterward, specimens were rinsed in running water, dehydrated in increasing concentrations of ethyl alcohol (Stanlab, Lublin, Poland), carried out through intermediate liquid Neo-Clear (Merck, Darmstadt, Germany), and embedded in paraffin blocks. The excessive paraffin was removed, exposing the surface of the sample with the embedded cardiac tissue. Subsequently, for every studied sample, four 600-nm-thick paraffin sections were prepared on a rotary microtome (Leica RM 2255, Nussloch, Germany). After deparaffinization, the sections were stained with Mayer's hematoxylin and eosin, HE (Bio-Optica Milano, Italy) and sealed with a Euparal mounting agent (Roth, Mannheim, Germany). Next, the histological slides have dried for seven days at 37°C. The evaluation of the severity of inflammation was performed for each studied group of animals: not immunized group without simvastatin, group with induced EAM without simvastatin, low-dosage simvastatin group EAM+S3, and high-dosage simvastatin group EAM+ S30. The assessment of each histological slide was conducted by two independent researchers, without knowledge about the way of the treatment of the animals. The mononuclear cell infiltrates (MCIs) were been assessed for the whole cardiac muscle section under a light microscope BX53 (Olympus, Tokyo, Japan) with magnification 20. Then, the inflammatory status in tissue sample was classified with the usage of the following scale developed by Godsel et al. [34]: normal (without any signs of inflammation); mild (≤10% of the whole cardiac tissue section was occupied by MCIs), moderate (>10% and ≤25% of the whole cardiac tissue section was occupied by MCIs), and severe (>25% of the whole cardiac tissue section was occupied by MCIs). The HE-stained cardiac tissue has been scanned with the usage of a Pannoramic Midi II histological scanner (3D HISTECH Ltd., Budapest, Hungary). The images were captured with the Pannoramic Viewer (3D HISTECH Ltd., RRID: SCR_014424, Budapest, Hungary) Software.“
Discussion is too short in evaluating similarities and differences to other possible use of statins as pleiotropic effect. In fact, evidences were already reported on the effects of statins during infections.
Response:
We thank Reviewer #2 for this comment. In response to Reviewer‘s comment the paragraph on page 11, lines 328–344 was changed as follows:
„In the present study, we found that in experimental autoimmune myocarditis in rats the use of a high dose of simvastatin (30 mg/kg/day) improved heart function and decreased degree of inflammation. These data correspond with several prior studies on statins in EAM [24-30]. The results of these studies, in which different statins were used (atorvastatin, fluvastatin, rosuvastatin, and pitavastatin), demonstrated the ability of this class of drugs to decrease severity of the disease and that effectiveness of statins was related to a sequential interference on the T-cell-mediated immune response [25-30]. Influence of simvastatin on EAM was investigated by Wu et al. They demonstrated that simvastatin ameliorated EAM through the inhibition of cross-talk between lymphocytes and antigen presenting cells [27].
In our study we demonstrated that high dose of simvastatin reduced mechanical dysfunction of heart muscle,decreased severity of inflammation along with attenuation of MMP-9 activity in comparison with low dose of simvastatin and decreased myofilaments degradation in hearts subjected to EAM. These data support the hypothesis that beneficial effects of simvastatin on EAM are associated not only with the previously reported influence on immune response, but also include a concomitant suppressive action on MMP-9 activity, leading to subsequent reduction in myofilaments degradation and prevention of mechanical dysfunction of the heart.“
Reviewer 3 Report
This paper is devoted to the study of simvastatin effect on the heart in experimental myocarditis in rats. The authors studied the effects of simvastatin on metalloproteinases activity, inflammation and contractile dysfunction. In most cases the manuscript is written well but there are some questions which need to be addressed.
The list of abbreviations used would be helpful for a reader.
In introduction section the authors should mention additional ways through which simvastatin could have its effect on miocarditis. The paper by Wu with coauthors (reference below) probably could help with this task.
Wu JL, Matsui S, Zong ZP, Nishikawa K, Sun BG, Katsuda S, Fu M. Amelioration of myocarditis by statin through inhibiting cross-talk between antigen presenting cells and lymphocytes in rats. J Mol Cell Cardiol. 2008 Jun;44(6):1023-31. doi: 10.1016/j.yjmcc.2008.03.016. Epub 2008 Apr 1. PMID: 18471827.
Materials and methods section is quite detailed.
Results section.
The authors should explain why they see an increase of MMP-9 activity in Figure 2 when they they applied low-dosage simvastatin in EAM group. It does not look like an inhibition of activity at all, also, even high dosage of simvastatin in EAM group does not cause a significant reduction of MMP-9 activity in comparison with just EAM group. Figure 2 is especially important since in serves as a significant basis for the name of the paper. Lines 295-298 in the Discussion section are also related to this question.
How was MMP-9 protein identified? Was it done just by molecular weight?
Figure 6 (as well as Figure 4) – it would be nice to have a representative images of control group which was not subjected to EAM.
In some rare cases there are grammatical errors which should be checked.
The paper can be accepted after minor revision related to answers to above mentioned questions/comments.
Author Response
Reviewer #3 comments:
This paper is devoted to the study of simvastatin effect on the heart in experimental myocarditis in rats. The authors studied the effects of simvastatin on metalloproteinases activity, inflammation and contractile dysfunction. In most cases the manuscript is written well but there are some questions which need to be addressed.
Response:
We thank the Reviewer #3 for positive comments and careful review, which helped to improve the manuscript. Please find below our answers to the specific comments.
The list of abbreviations used would be helpful for a reader.
Response:
As per Reviewer #2 suggestion the list of abbreviations was prepared as follows and incorporated into the manuscript on pages 11–12, line 365–385:
„Abbreviations:
MMPs – matrix metalloproteinases
EAM – experimental autoimmune myocarditis
MMP-9 – matrix metalloproteinase 9
MMP-2 – matrix metalloproteinase 2
MMP-3 – matrix metalloproteinase 3
SARS-CoV-2 – severe acute respiratory syndrome coronavirus 2
HMG-CoA – 3-hydroxy-3-methylglutaryl coenzyme-A
I/Ri – ischemia-reperfusion injury
TNF-alpha – tumor necrosis factor alpha
IFN-gamma – interferon gamma
LV – left ventricular
IVSDD – interventricular septum diameter
LVEDD – left ventricular end-diastolic diameter
LVESD – left ventricular end-systolic diameter
LVPWDD – left ventricular posterior wall end-diastolic diameter
FS – fractional shortening
TEM – transmission electron microscopy
HE – hematoxylin/eosin
SDS-PAGE – sodium dodecyl sulphate-protein acrylamide gel electrophoresis
AU – arbitrary unit”
In introduction section the authors should mention additional ways through which simvastatin could have its effect on miocarditis. The paper by Wu with coauthors (reference below) probably could help with this task.
Wu JL, Matsui S, Zong ZP, Nishikawa K, Sun BG, Katsuda S, Fu M. Amelioration of myocarditis by statin through inhibiting cross-talk between antigen presenting cells and lymphocytes in rats. J Mol Cell Cardiol. 2008 Jun;44(6):1023-31. doi: 10.1016/j.yjmcc.2008.03.016. Epub 2008 Apr 1. PMID: 18471827.
Response:
As per Reviewer #3 suggestion the paper by Wu et al. was cited in the text on page 2, lines 68 and 70, and added to reference list on page 14, line 477, and the following sentence was incorporated into the text:
„The results of many studies demonstrated the ability of statins to decrease both histopathological severity of myocarditis and improve cardiac function through interference on the T-cell-mediated immune response [24-29].“
Materials and methods section is quite detailed
Response:
We thank the Reviewer #3 for this positive comment.
Results section.
The authors should explain why they see an increase of MMP-9 activity in Figure 2 when they they applied low-dosage simvastatin in EAM group. It does not look like an inhibition of activity at all, also, even high dosage of simvastatin in EAM group does not cause a significant reduction of MMP-9 activity in comparison with just EAM group. Figure 2 is especially important since in serves as a significant basis for the name of the paper. Lines 295-298 in the Discussion section are also related to this question.
Response:
We thank the Reviewer #3 for this comment. Indeed, the paragraphs were changed as follows (page 6, lines 224–228 and page 11 lines 338–340:
„As a result of EAM, there was an enhanced activation of MMP-9 especially in EAM+S3 group, which was abolished by high dose of simvastatin – it was a significant reduction of MMP-9 activity in the EAM+S30 group in comparison with the EAM+S3 group (58.37±33.63 AU/μg vs 10.30±8.72 AU/μg, n=5 p< 0.05) (Fig. 2).“
„In our study we demonstrated that high dose of simvastatin reduced mechanical dysfunction of heart muscle,decreased severity of inflammation along with attenuation of MMP-9 activity in comparison with low dose of simvastatin and decreased myofilaments degradation in hearts subjected to EAM.“
How was MMP-9 protein identified? Was it done just by molecular weight?
Response:
In response to Reviewer #3 question we would like to expalin, that MMPs were identified on the basis of their molecular weight in comparison to a capillary blood standard prepared according to Makowski and Ramsby. Moreover, during method validation for each type of material we perform preliminary experiments in which gels are incubated with the addition of o-phenantroline as a non-specific inhibitor of MMPs.
The fallowing sentence was added to parapraph „2.5. Measurement of MMP-2 and MMP-9 by gelatine zymography“ on page 4, lines 142–144:
„MMPs were identified on the basis of their molecular weight in comparison to a capillary blood standard prepared according to Makowski and Ramsby [33].“
Incorporating this sentence to the paragraph required the addition of 1 new citation which is now added to reference list on page 14, line 493–494.
At the same time we decided to explain in more detail preparation of the sample for zymography and western blotting. The following text was added to paragraph „2.4. Sample collection“ on page 3, line 120–127:
„Prior to zymography and western blotting a proper amount of heart powder was mixed with homogenization buffer (50 mmol/L Tris-HCl pH 7.4, 150 mmol/L NaCl, 0.1% Triton X-100, and Protease Inhibitors Cocktail Set III (Sigma-Aldrich) to prepare 20% homogenates (w/v) with subsequent mechanical homogenization (3 times for 10 seconds on ice) and centrifugation (10 000 × g, for 5 min at 4°C). Protein content in supernatants was measured with Bradford Protein Assay (Bio-Rad, Warszawa, Poland) and bovine serum albumin (heat shock fraction, ≥ 98%, Sigma-Aldrich) served as a protein standard.“
Figure 6 (as well as Figure 4) – it would be nice to have a representative images of control group which was not subjected to EAM.
Response:
We are very grateful for this valuable comment. The images of the heart of the control group without inflammation were presented in the new figure with an adequate description (page 7–8, lines 251–254):
“Figure 4. The representative images of cardiac muscle from not immunized rats without simvastatin (group C). The healthy heart without mononuclear cell infiltrates (A, HE stain) and a typical ultrastructure of heart with proper cross striations and abundant mitochondria (B, TEM).”
The numbers of subsequent figures needed to be changed as well as citations numbers in the text (page 8, lines 263–266; page 8, line 270, page 9 lines 304, page 9 lines 271–272, line 278; page 9 line 286; page 10 lines 287–288).
Also two new sentences were incorporated to the text on page 7, line 250 and page 9, lines 277–278:
„There was no inflammation observed in the control group (group C) (Fig. 4).“
„The cardiac muscle in the not immunized rats demonstrated normal cross striations and ultrastructure of myofibrils with well-preserved, plentiful mitochondria (Fig. 4).“
In some rare cases there are grammatical errors which should be checked.
Response:
In response to Reviewer #3 comment the text undewent English revision.
The paper can be accepted after minor revision related to answers to above mentioned questions/comments.
Response:
We greatly appreciate this comment. Changes suggested have been done and we do hope that the manuscript is now acceptable for publication in Biomolecules, Special Issue "Matrix Metalloproteinases in Health and Disease 2.0"
Round 2
Reviewer 1 Report
present in present form
Author Response
/
Reviewer 2 Report
The authors reply to all required issue. They made a good manuscript and a good research.
Author Response
/